# The Morphology of Cross-Beaks and *BMP4* Gene Expression in Huiyang Bearded Chickens

**DOI:** 10.3390/ani9121143

**Published:** 2019-12-13

**Authors:** Yuyu Hong, Yuchang Pang, Haiquan Zhao, Siyu Chen, Shuwen Tan, Hai Xiang, Hui Yu, Hua Li

**Affiliations:** 1Guangdong Provincial Key Laboratory of Animal Molecular Design and Precise Breeding, Foshan University, Foshan 528225, China; 2111759007@fosu.edu.cn (Y.H.); pangyuchang2018@163.com (Y.P.); fszhaohq@163.com (H.Z.); jihebuluo2015@163.com (S.C.); tsw_gogo@hotmail.com (S.T.); vamyluo@126.com (H.X.); yu71hui@aliyun.com (H.Y.); 2Xianxi Biotechnology Co. Ltd, Foshan 528200, China

**Keywords:** Huiyang bearded chickens, cross-beaks, *BMP4* gene, craniofacial bones

## Abstract

**Simple Summary:**

Recently, the emergence of cross-beaks has been reported in several domestic chickens. Despite several candidate genes, *bone morphogenetic protein 4* (*BMP4*) has been suggested as responsible for chicken cross-beaks. The subtypes of the morphology term, etiopathogenesis, and the relationship of the candidate *BMP4* gene to cross-beaks are not yet known. The objective of this study was to describe the subtypes of cross-beaks by left or right and upper and lower jaw bones and to figure out the relationship between *BMP4* and the development of craniofacial bones in Huiyang bearded chickens.

**Abstract:**

Bird beaks are important for biological purposes such as food intake, removing parasites, and defining phenotypic attributes. Cross-beaks are a threat to poultry health and are harmful to productivity, wasting some units in the poultry industry. However, there is still limited research on subtypes of cross-beaks and the genetic basis of cross-beaks as well. Here, we described the subtypes of cross-beaks in terms of left or right and upper or lower jaw bones. We evaluated the impact of cross-beaks on craniofacial bones and figured out the relationship between *bone morphogenetic protein 4 (BMP4)* and the development of craniofacial bones in Huiyang bearded chickens. We identified five typical subtypes of cross-beaks by morphological assessment and X-ray scanning. We found that cross-beaks caused certain changes in the facial bone morphology, including changes to the length and width of the bone around the ocular area (*p* < 0.05). The relative expressions of *BMP4* in lacrimal, mandible, premaxilla, frontal, and parietal bones were significantly higher in the severe cross-beak group, followed by that of the medium cross-beak group, weak cross-beak group, and control group (*p* < 0.05). Overall, we constructed a generally applicable method to classify cross-beaks in term of the angle. The skeleton around the ocular area was affected by the cross-beak. The expression levels of *BMP4* in craniofacial bones may provide insight to potential role of *BMP4* in the development of cross-beaks.

## 1. Introduction

A cross-beak is a dysmorphology relating to deformed beaks and has been suggested as a threat to animal welfare breeding. There are 1%–3% of Chinese native chickens that have beak deformities, leading to great economic loss and animal welfare issues in the poultry industry [1,2,3]. Typically, cross-beaks refer to the misalignment of upper and lower beaks. Birds afflicted by this dysmorphology have a decreased ability to intake feed and drink and, consequently, have a higher susceptibility to parasites and pathogens and an even much higher mortality. Therefore, it is of importance to further understand the morphology and molecular etiopathogenesis of deformed beaks in order to reduce economic loss.

Cross-beaks have been identified as four different types by Landauer [4]. The first two categories indicate that the cross-beak is accompanied by abnormalities of eyes and skulls, respectively. The development of a cross-beak within the first two months of age has been suggested as the third type, while the fourth category describes the presence of a cross-beak at hatching but that later grows into a normally developed beak. Despite the above classification, there is still lack of detailed descriptions for subtypes of cross-beaks, which will give us a better understanding of beak deformity in chickens.

Various factors including trauma, abrasion of the rhamphotheca, exposure to toxins [5,6], nutritional deficiencies [7], and hypoxia [8] have been proposed to contribute to beak deformity in birds and chickens. Noteworthily, genetics are an essential factor [9]. To date, more than 20 candidate genes, including the *bone morphogenetic protein* gene family (*BMP*), related to cross-beaks have been proposed in chickens [1,9,10]. As a member of the *BMP* gene family and TGF-beta superfamily, *bone morphogenetic protein 4* (*BMP4*) is involved in several processes of skeletal development, such as cranio-maxillofacial bone formation [11,12], and it is deemed as a critical candidate gene for beak formation and evolution [13,14]. Previous reports demonstrated that *BMP4* plays a critical role in craniofacial traits [13,15,16]. Furthermore, a digital gene expression (DGE) profile proposed that *BMP4* might serve in beak development, but the authors did not mention the relationship of *BMP4* with cross-beaks [1]. Therefore, whether *BMP4* is involved in the development of cross-beaks is not yet known.

In this study, we first constructed the nomenclature of cross-beaks by the left or right and upper or lower deflection of the jaw bones, then we defined the morphology and graded the angle of the cross-beak. Afterward, we determined the expression levels of *BMP4* in craniofacial bones of cross-beaks with different crossed angles. We aimed to provide insights into the underlying genetic mechanism of *BMP4* in craniofacial bones in the development of chicken cross-beaks. 

## 2. Materials and Methods 

### 2.1. Animals 

The study was approved by the Animal Care Committee of Foshan University (Foshan, People’s Republic of China; Approval ID: FOSU#056). All animal experiments were performed according to national and international guidelines for animal welfare. A total of 259 out of 6276 chickens, including 150 abnormal chickens as case group and 109 normal chickens as control group, were selected for the study. All chickens were collected from one pure line of Huiyang bearded chickens and fed with corn–soybean-type diets at Foshan University.

### 2.2. Experimental Design

#### Classification of a Cross-Beak

All chickens that were characterized as having a cross-beak, by one or both beaks deviating laterally from the longitudinal axis of the head along with the backbone, were selected according to a previous study [3]. The main manifestation is that the upper and lower beaks are dislocated and cannot be properly closed. Centered on the spine and the midline of the face, we constructed the nomenclature of the cross-beak. If the lower beak was normal and the upper beak deviated to the left or right, is was accordingly named the left-upper or right-upper cross-beak. Likewise, if the upper beak was normal and the lower beak deviated to the left or right, it was called the left-lower or right-lower cross-beak. When both the upper and lower beaks were bias to the midline axis, and deflected in opposite directions, this was named polarization cross-beak. X-ray (LX-24HA, Lang’an Tec. Co. Led., Beijing, China) was used to observe the morphological changes of cross-beaks. 

### 2.3. Samplings

#### 2.3.1. Morphology

All experimental chickens of 105 d of age were slaughtered by rapid decapitation. Afterwards, the deviated angle of a cross-beak was immediately measured by a protractor before necropsy. We defined three cross-beak groups in terms of their deviated angles. The angles of weak, medium, and severe cross-beak chickens were 1°–10°, 10°–20°, and ≥20°, respectively.

The distance of the left/right eye midpoint to the upper beak (DLU/DRU), the length of left/right ocular (LOL/ROL) and width of the left/right ocular (LOW/ROW), and that of the left/right eye midpoint to the nostril (DLN/DRN) were determined with a Vernier caliper. Thereafter, the infraorbital sinus was exposed by cutting along the yellow dotted line (Figure 1). The lengths of the left/right infraorbital sinus (LIS/RIS) were measured with a Vernier caliper (Figure 1).

#### 2.3.2. Gene Expressions 

When slaughtered and dissected, the tissues of lacrimal, turbinate, premaxilla, nasal, mandible, parietal, frontal, and occipital bones were independently collected from six female chickens in each group (weak, medium, severe, and control). The samples were immediately frozen in liquid nitrogen and then stored at −80 °C until RNA isolation. The bone tissues were placed in a beaded tube (SC Micro Tube, Sarstedt) with Trizol reagent (Invitrogen, Massachusetts, USA). FastPrep-24™5G (MP Biomedicals) was used to agitate the sample tube at a speed of 5 m/s for 35 s. Then, samples were centrifuged at 12,000 rpm for 15 min at 4 °C, and the upper aqueous layer was transferred to a new tube for the next RNA extraction. The residual gDNA and protein were removed with Dnase I (TaKaRa, Dalian, China) and RNA clean kit (TIANGEN, Beijing, China). The purified RNA was dissolved (200–400 ng/mL, OD260/OD280 = 1.8–2.0) and stored at −80 °C. After that, 2 μL of total RNA and 3 μL of Oligo(dT)_18_ were mixed at 70 °C for denaturation and immediately placed on ice for 5 min. Then, 7.5 μL of ddH_2_O, 4 μL of Reverse Transcription 5× Buffer, 2 μL of dNTPs (10 mM), 0.5 μL of Recombinant RNasin^®^ Ribonuclease Inhibitor, and 1 μL of M-MLV Reverse Transcriptase were added in each 20 μL Reverse Transcription reaction system. After being fully mixed, they underwent reverse transcription at 42 °C for 1 h and finally inactivated to reverse transcriptase at 72 °C for 10 min. The cDNA was stored at −80 °C for subsequent qRT-PCR.

Then, qRT-PCR was performed to determine the expression of *BMP4* in different skull parts using the ABI 7500 Realtime Detection System (Applied Biosystems, Massachusetts, USA) and RT-PCR reagents (TransGen Biotech, Beijing, China). Each 20 μL PCR reaction system contained 10 μL of 2 × TransStart Top/Tip Green Qpce, 0.4 μL (10 pM) of each primer, 0.4 μL of Passive Reference Dye (50×), 0.8 μL of cDNA (100 ng), and 8 μL of ddH2O. After an initial denaturing for 30 s at 95 °C, there were 40 cycles of amplification (95 °C for 15 s, 57 °C for 30 s, and 72 °C for 85 s), followed by thermal denaturing to generate melting curves. *GAPDH* was amplified in the same plates as endogenous controls. Samples were assayed in triplicate for standard curves. PCR efficiency, amplification efficiency of the transcripts of interest, and the internal standard of *GAPDH* were consistent with the measurement of *BMP4*. Dissociation curves verified that amplification was specific. Relative quantitative expression of the target gene was calculated using the 2^−ΔΔCt^ method [17]. The primers of *BMP4* and *GAPDH* are shown in Table 1.

### 2.4. Statistical Analyses

The data for angle, distance, and gene expression were expressed as mean ± standard deviation (M ± SD). We first analyzed the difference of gender (proportion) among left-crossed chickens, right-crossed chickens, and normal chickens by using chi square tests. Since the gender did not affect the type of crossed beak, we then analyzed the difference (proportion) of left-upper cross-beak, left-lower cross-beak, right-upper cross-beak, right-lower cross-beak, and polarization cross-beak chickens as well as the upper cross-beak and lower cross-beak chickens regardless of gender by a chi square test. Data of gene expression and distance were checked for normality and homogeneity of variance and were transformed where necessary. Samples for gene expression were selected for female chickens, which were analyzed using one-way (control, weak, medium, and severe) analysis of variance (ANOVA) by SPSS 20.0. While, data of distance were analyzed using two-way analysis of variance (ANOVA) by SPSS 20.0. The gender and treatment (right-crossed beak, left-crossed beak, and normal beak; control, weak, medium, and severe) as factors were included in the model. Yijk = μ + Ai + Bj + (AB)ij + εijk, where Yijk is observation k (angles/distances) at the level i type of beak (TOB) and the level j gender (G), μ is the overall mean, Ai is the effect of level i type of beak, Bj is the effect of level j gender, (AB)ij is the effect of the interaction of level i TOB with level j G, and εijk is the random error with mean 0 and variance σ^2^. Significance tests were analyzed using Duncan’s multiple range test, and the effects and differences were considered statistically significant when *p* < 0.05.

## 3. Results

### 3.1. Morphological Observations and Descriptions of the Five Subtypes of Cross-Beaks

Referring to the above nomenclature, there were five subtypes of cross-beaks versus the normal beaks (Figure 2A). They were the left-upper cross-beak (Figure 2B), left-lower cross-beak (Figure 2C), right-upper cross-beak (Figure 2D), right-lower cross-beak (Figure 2E), and the polarization cross-beak (Figure 2F). Particularly, the number of deformed beaks included 31 left-upper cross-beaks (hens: 19; roosters: 12; 20.67%), 32 left-lower cross-beaks (hens: 18; roosters: 14; 21.33%), 16 right-upper cross-beaks (hens: 5; roosters: 11; 10.66%), 50 right-lower cross-beaks (hens: 38; roosters: 12; 33.34%), and 21 polarization cross-beaks (hens: 15; roosters: 6; 14%), respectively. Gender did not show effects on those subtypes of deformed beaks. The occurrence rates of left (42%) and right (46%) cross-beaks did not differ, but both of them were higher than the rate of polarization cross-beaks (*p* < 0.05). Meanwhile, lower cross-beaks (54.67%) occurred more frequently than upper cross-beaks (31.33%) (*p* < 0.05).

The average angle of left cross-beaks was 9.86° (Table 2). Among them, the left-upper cross-beaks was 11°, ranging from 2° to 31°, and the left-lower cross-beaks was 8.59°, ranging from 2° to 20°. The average angle of the right cross-beaks was 9.78° (Table 2). Among them, the right-upper cross-beaks was 12.23°, and the variation range was from 5° to 45°, while that of the right-lower cross-beaks was 9°, and the variation range was from 2° to 25°. The average angle of the polarization cross-beaks was 12.24°, ranging from 6° to 26° (Table 2). In all 150 cases, weak, medium, and severe cross-beak chickens accounted for 67.3%, 25.33%, and 7.37% of the total, respectively. In addition, except for three times as many hens crossing on the right-lower cross-beak as there were roosters, the ratio of males to females was basically 1:1 in the other four subtypes. All the subtypes were independent of gender (Table 2). 

### 3.2. The Effect of Cross-Beaks on Chicken Faces 

The gender and the interaction of gender × TOB showed no effects on DLU (Figure 3A), DRU (Figure 3B), DRN (Figure 3D), LOW (Figure 3G), ROW (Figure 3H), LIS (Figure 3I), and RIS (Figure 3J). The DLU and DLN were affected by the TOB, with shorter distances in LC chickens than RC and NC chickens (*p* < 0.01, Figure 3A,B). Whereas, the DLN in female chickens with left cross-beaks was shorter than male chickens in left cross-beak chickens (*p* < 0.05), but not in RC and NC chickens (Figure 3C). The DRU and DRN were also affected by the TOB, with shorter values in right-crossed beak chickens than left-crossed beak and normal chickens (*p* < 0.01, Figure 3B,D). The LOL was affected by the TOB (*p* < 0.01) and G (*p* < 0.01) (Figure 3E), showing a longer distance in right-crossed chickens followed by left-crossed chickens and normal chickens (*p* < 0.01). Besides, the LOL in female chickens with left/right cross-beaks was shorter than male chickens with left/right cross-beaks (*p* < 0.01). The ROL was affected by the TOB (*p* < 0.01) and G (*p* < 0.05) (Figure 3F), with a longer distance in left-crossed chickens than right-crossed chickens and normal chickens (*p* < 0.01). Furthermore, that distance in female chickens with left/right cross-beaks was shorter than male chickens with left/right cross-beaks (*p* < 0.05). The LOW and ROW were affected by TOB, with short values in chickens with left/right cross-beaks than normal chickens (*p* < 0.01, Figure 3G,H). The ROW was affected by TOB (*p* < 0.01) (Figure 3H). In chickens with left/right cross-beaks, the ROW was shorter than normal chickens (*p* < 0.01). The LIS and RIS were affected by TOB (*p* < 0.01) (Figure 3I,J). The LIS was longer in right cross-beaks followed by normal chickens and the left cross-beaks (*p* < 0.01). The LIS was longer in left cross-beaks than normal chickens and right crossed beak chickens (*p* < 0.01).

### 3.3. The Relative Expression of BMP4 in Craniofacial Bone 

The relative expressions of *BMP4* in mandible, premaxilla, lacrimal, frontal, and parietal bones were significantly higher in the severe cross-beak group, followed by that of the medium cross-beak group, weak cross-beak group, and control group (*p* < 0.01) (Figure 4A). But there were no significant differences in nasal, turbinate, and occipital bones among groups (Figure 4B).

## 4. Discussion 

Cross-beaks are a disease resulting in economic loss for the poultry industry and comprise 1% to 3% in Chinese local chickens [2]. A beak consists of multiple facial prominences. Each prominence has a distinct shape, and all of them are coordinated at proportional sizes to compose a specific beak [18,19]. The skull is formed from the neurocranium and viscerocranium. The neurocranium includes the frontal, parietal, occipital, and sphenoid bones, while the viscerocranium consists of multiple facial bones such as the mandible, maxilla, nasal, and so on [20]. The upper and lower jaw bones are part of the craniofacial bone tissue, and any changes to them may shape the development of a beak. In this study, we observed significant deflections of the maxilla and mandible, which were consistent with previous studies [3,5]. In Appenzeller Barthuhn chickens, cross-beaks were more likely to deflect to the right side [3]. Even though no different occurrence was observed between the left and right cross-beaks in this study, lower cross-beaks were found to be more likely to deflect to the right side. Moreover, lower cross-beaks occurred much more than upper cross-beaks. These findings indicate that the lower beak is more prone to deformities, which may be supported by a previous study describing that changes to the mandible are underneath changes to the facial bones, and the mandible moves more easily than other facial bones [21,22]. Additionally, the deviation of angles (more than half of chickens with cross-beaks) were mainly concentrated at 1°–10°, which is similar with the above study that had 50% of all deviation angles between 1° and 9° for the upper beak and 2° to 9° for the lower beak [3].

Since severe cross-beaks influence food and water intake and survival, some chickens may not grow to the adult age. Thus, it makes sense that weak cross-beak chickens show greater proportions among the chicken population with deformed beaks. We did not find that the upper and lower beaks deflected to the same direction simultaneously, regardless of what was previously reported [3]. In cross-beak chickens, the LOL/ROL were significantly longer than that of the normal chickens, indicating that the periorbital area of the cross-beak chickens is affected occasionally, and some cross-beaks may have occurred in the bone tissue around the ocular area. Similar results have been observed in humans: the cranial base angle affects the anteroposterior orientations of both the maxilla and mandible, with a slight contribution to the appearance of different vertical skeletal patterns, and the cranial base length is linked to total facial height, which can lead to the appearance of different vertical skeletal patterns [23]. Recent research about the modern human mandible during ontogeny concluded that the mandible has maintained a passive role in hominin skull evolution, playing ‘‘follow the leader’’ with the cranium [22,24]. Furthermore, considering cross-beaks do not show a gender effect, it may belong to incomplete dominant inheritance in autosomes. Up to now, it is very difficult to get the major gene of incomplete dominant inheritance.

Although most of the cross-beak lesions were observed to change in the upper and lower jawbones, our morphological results indicate that the occurrence of deformed beaks is likely related to the malformed development of craniofacial bones. The external differences in beak morphology reflect differences in their respective craniofacial skeletons [25]. Thus, cross-beaks are probably caused by left–right asymmetric growth of the beaks during development, which is probably linked to the development of the craniofacial bone. As known, the *BMP4* is a critical gene for the development of the beak [26,27]. Our results further found the potential connection of *BMP4* with cross-beaks. Its expression in multiple craniofacial bones suggested that *BMP4* might be involved in the regulation of multiple craniofacial bone growth and development pathways in chickens, such as the development of lacrimal, mandible and the premaxilla, parietal, and frontal bones. Previous studies indicated that overdosed *BMP* activities were also detrimental to multiple axial skeletal tissues and facial skeletal formation [26,27,28]. In the development of eyes, increased levels of *BMP4* signaling caused decreased proliferation, reduced retinal volume, and altered the shape of the optic cup [29]. Moreover, *BMP4,* with malformations of the skull bone, tissues, and organs, is species-specific. For example, in blind cave-dwelling fish, *BMP4* was associated with asymmetric craniofacial development [30]. In humans, elevated *BMP* signals disrupted the normal development of the palate and teeth [31,32]. Other studies demonstrated that *BMP4* is related to the malformation of tissues and organs, such as dysplasia of the lumbosacral spinal cord in rats [33], in multiplex nonsyndromic cleft lip with or without cleft palate in humans [34,35], and in enhancing astrocytosis, microgliosis, and amyotrophic lateral sclerosis [36]. Furthermore, the ectopic osteogenesis of fibrodysplasia ossificans progressiva is associated with overexpression of *BMP4* in lymphocytes, which suggests that an inappropriate enhancement of *BMP4* expression may play an essential role in the molecular pathophysiology of fibrodysplasia ossificans progressiva [37,38]. Specifically, our data suggested that, in chickens, *BMP4* might be linked to alternations in parietal, frontal, lacrimal, mandible, and premaxilla bones, but not in turbinate, nasal, and occipital bones. Intriguingly, the level of *BMP4* along the entire dorso-ventral axis of the optic cup suggests that the activity of *BMP4* is not limited by distance [29]. It seems to account for our results that the high expression of *BMP4* in bones at a certain distance from the beak is the cause of the occurrence of cross-beaks.

In this study, the relative expression of *BMP4* was the highest in the severe cross-beak chickens, as expected, and followed by medium cross-beak chickens, weak cross-beak chickens, and normal-beak chickens in sequence. This finding was identical to the previous report, that an increase in the deviation angle of the beak was accompanied with an increase in *BMP4* administration, and could induce the phenotype similar to that of beak deformity [39]. Noteworthily, the phenotype changes of a beak vary by different injection sites and doses of *BMP4*, for instance, the injection of exogenous *BMP4* into the neural crest-derived interstitium resulting in an increase in width and depth of the beak [13].

In addition, overdosed *BMP* signaling results in dramatic apoptosis in cranial neural crest (CNC) cells and their derivatives [26,40,41]. Lineage tracing revealed that most of the facial skeleton is derived from CNC cells [42,43]. The specification, emigration and migration, proliferation, survival, and ultimate fate of the CNC plays an important role in regulating craniofacial development [44]. Besides, CNC was found to underly the functional morphology and evolution of the jaw by modulating the activity of mesoderm-derived osteoclasts and bone resorption [45]. These provide further insights into the molecular mechanism of cross-beaks.

## 5. Conclusions

There are five subtypes of cross-beaks in Huiyang bearded chicken. The skeleton around the ocular area is affected by a cross-beak. The over-expressions of *BMP4* in parietal, frontal, lacrimal, mandible, and premaxilla bones result in cross-beaks, but not in turbinate, nasal, and occipital bones. This finding provides insight on the potential role of *BMP4* in the development of cross-beaks. 

## Figures and Tables

**Figure 1 animals-09-01143-f001:**
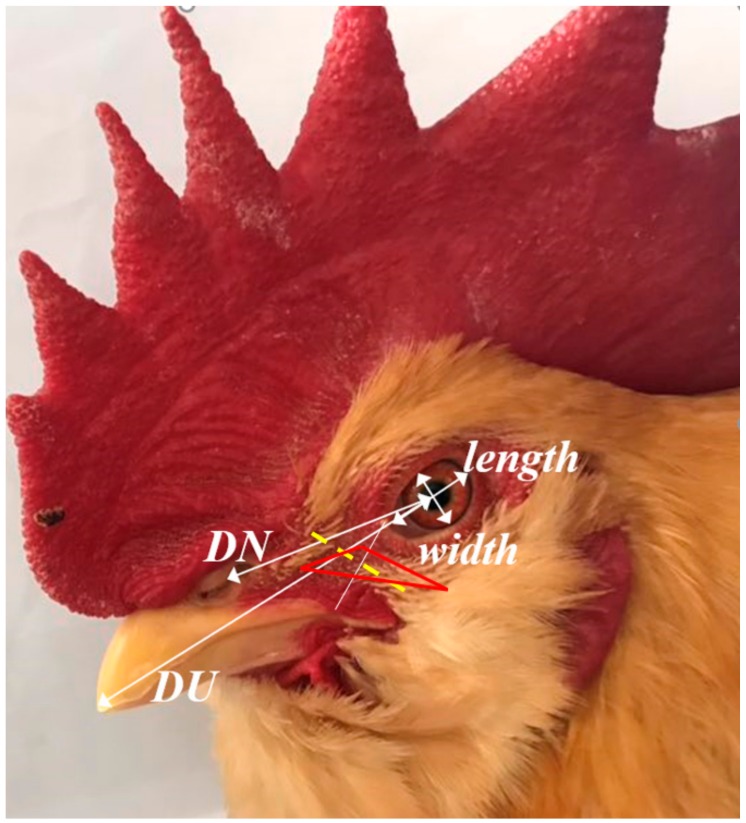
Schematic diagram of facial measurements. DU: distance of the left/right eye midpoint to the upper beak; DN: distance of the left/right eye midpoint to the nostril. The red region is the location of the infraorbital sinus. The yellow dotted line is vertically divided equally between the line connecting the leading edge of the corner of the eye and the commissure of lips.

**Figure 2 animals-09-01143-f002:**
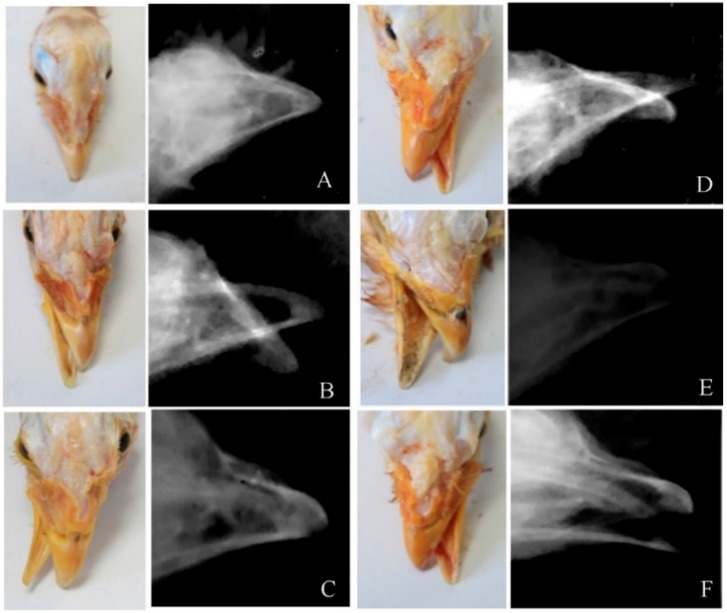
The morphology of cross-beaks in Huiyang bearded chickens. (**A**): normal beak; (**B**): left-upper cross-beak; (**C**): left-lower cross-beak; (**D**): right-upper cross-beak; (**E**): right-lower cross-beak; and (**F**): polarization cross-beak.

**Figure 3 animals-09-01143-f003:**
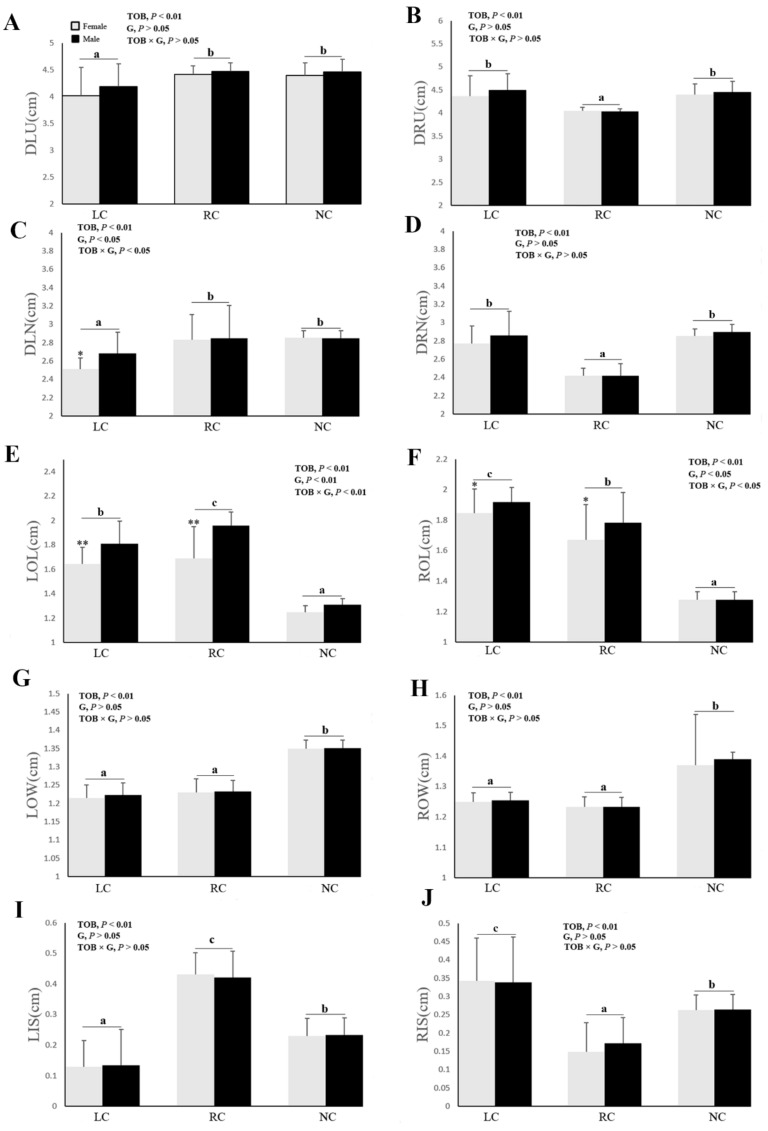
Comparison of the craniofacial bone morphology (mean ± SD) among the left cross-beak, right cross-beak, and normal chickens including female and male individuals. (**A**) DLU: distance of the left eye midpoint to the upper beak; (**B**) DRU: distance of the right eye midpoint to the upper beak; (**C**) DLN: distance of the left eye midpoint to the nostril; (**D**) DRN: distance of the right eye midpoint to the nostril; (**E**) LOL: left ocular length; (**F**) ROL: right ocular length; (**G**) LOW: left ocular width; (**H**) ROW: right ocular width; (**I**) LIS: left infraorbital sinus; (**J**) RIS: right infraorbital sinus. LC: Left cross-beak (left-upper cross-beak + left-lower cross-beak, n = 63); RC: right cross-beak (right-upper cross-beak + right-lower cross-beak, n = 66); NC: normal beak (n = 109). TOB means type of beak, G means difference of gender. * *p* < 0.05, ** *p* < 0.01 means significant difference between gender. Values with different small letters (a, b, c) indicate statistically significant differences (*p* < 0.05).

**Figure 4 animals-09-01143-f004:**
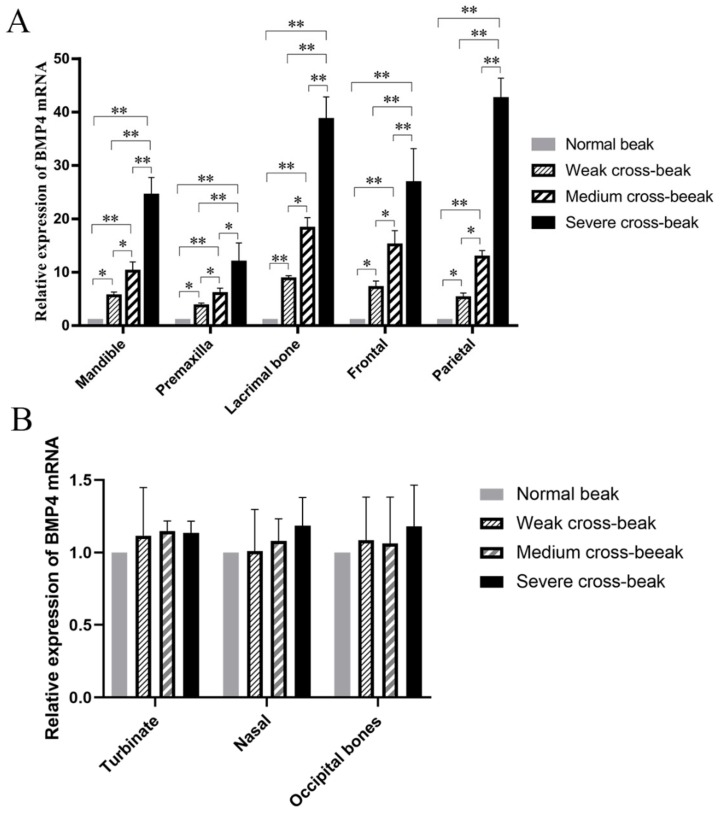
The relative expression levels of *BMP4* in craniofacial bones. (**A**) The relative expression levels of *BMP4* in mandible, premaxilla, lacrimal, frontal, and parietal bones. (**B**) The relative expression levels of *BMP4* in nasal, turbinate, and occipital bones. * *p* < 0.05, ** *p* < 0.01 among the groups.

**Table 1 animals-09-01143-t001:** The primer information of *BMP4* and *GAPDH*.

Gene	Primer Sequence (5′→3′)	Temperature (°C)	Product Length (bp)
*BMP4*	F: AGCATCCCCAACATCCAGAA	59	233
	R: CAGAACTTGGAGGGCTGGTA		
*GAPDH*	F: CCTCTCTGGCAAAGTCCAAG	57	200
	R: CATCTGCCCATTTGATGTTG		

**Table 2 animals-09-01143-t002:** Basic information of the five beak subtypes.

Type	Subtypes	Mean ± SD, Range (°)	No. ^1^	Proportion (%)
Left cross-beak	Upper	11.00 ± 8.04, (2–31)	31	20.67
	Lower	8.59 ± 5.10, (2–20)	32	21.33
Right cross-beak	Upper	12.23 ± 10.50, (2–45)	16	10.66
	Lower	9.00 ± 6.02, (2–25)	50	33.34
Polarization cross-beak	Polarization	12.24 ± 4.30, (6–26)	21	14

^1^ The number of hens and roosters in each type of chicken, of which the former data are the number of hens and the latter are that of roosters.

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
