# Peer review of "The Morphology of Cross-Beaks and BMP4 Gene Expression in Huiyang Bearded Chickens"

_animals, 2019, doi:10.3390/ani9121143_

Round 1
Reviewer 1 Report
There are some issues should be addressed in the pdf file before the manuscript publication

Author Response
Dear Editors and Reviewers:
Thank you for your letter and for the reviewers’ comments concerning our manuscript entitled “animals-631631”. Those comments are all valuable and very helpful for revising and improving our paper, as well as the important guiding significance to our researches. We have studied comments carefully and have made correction which we hope meet with approval. Revised portion are marked in red in the paper. The main corrections in the paper and the responds to the reviewer’s comments are as flowing:
Responds to the Reviewers’ comments:
Point 1:Line 18-19, “As a tactile sensory organ, the bird beak is important for the intake of food and water, and for grooming purposes as well as parasite removal.” Replace to “The bird beak is important for a biological purposes such as food intake, parasite removing, and phenotypic attributes.”
Response 1: Thank you! According to your suggestion, we have revised it in the line 19-20.
Point 2: Line 19-20, Changes were made here to improve the clarity and readability of this part. Please check whether the revised part retains the intended meaning.
“The cross-beak is a threat to poultry health and harmful effect for its productivity, wasting some units in the poultry industry.”
Response 2: Thank you for your nice suggestion! We have made modifications in the line 20-21.
Point 3: Line 21, Replace “lack” to “limited”
Response 3: Thank you! According to your suggestion, we have replaced “lack” to “limited” in the line 21.
Point 4: Line 23, Replace “bones, evaluate” to “bones and evaluated”
Response 4: Thank you! According to your suggestion, we have replaced “bones, evaluate” to “bones and evaluated” in the line 23.
Point 5: Line 25, Replace “morphology presentation” to “morphological assessment”
Response 5: Thank you! According to your suggestion, we have replaced “morphology presentation” to “morphological assessment” in the line 26.
Point 6: Line 27, Please provide the expansion, if any, for the abbreviation “BMP4” in the abstract and also in its first occurrence wthin the main text.
Response 6: We are very sorry that we have neglected to provide gene’s full name in first instance of its abbreviation. According to your suggestion, we have supplemented the expansion of “BMP4” in the line 12, line 24 and the line 58.
Point 7: Line 53-54, “Indeed, a few studies were conducted on certain species using the method of genome-wide association studies (GWAS)” delete, out of scope.
Response 7: Thank you! According to your suggestion, we have deleted it in the line 55.
Point 8: Line 70, Please provide the ethics statement.
Response 8: We are very sorry for ignoring the addition of ethics statement due to our mistake. We have added “The study was approved by the Animal Care Committee of Foshan University (Foshan, People’s Republic of China; Approval ID: FOSU#056). All animal experiments were performed according to national and international guidelines for animal welfare.” in the line 72-74.
Point 9: Line 76-78, “The cross-beak is characterized by one or both beaks deviating laterally from the longitudinal axis of the head along with backbone [3]. The main manifestation is that the upper and lower beaks are dislocated and cannot be properly closed.” replace to “All chickens which characterize of cross-beak by one or both beaks deviating laterally from the longitudinal axis of the head along with backbone were be selected according to [3]. The main manifestation is that the upper and lower beaks are dislocated and cannot be properly closed.”
Response 9: Thank you! According to your suggestion, we have revised it in the line 80-83.
Point 10: Line 79-85, “Centered on the spine and the midline of the face, we constructed the nomenclature of the cross-beak. If the lower beak is normal, the upper beak deviation to the left or right is accordingly named after the left-upper or right-upper cross-beak. Likewise, if the upper beak is normal, the lower beak deviation to the left or right is called the left-lower or right-lower cross-beak. When both the upper and lower beaks are bias to the midline axis, and deflect opposite directions, namely polarization cross-beak. X-ray (LX-24HA, Lang’an Tec.Co.Led., Beijing) was used to observe the morphological changes of cross-beaks.” merge with the previous paragraph.
Response 10: Thank you! According to your suggestion, we have merged it with the previous paragraph. In the line 83.
Point 11: Line 99-100, “The samples were flash frozen in liquid nitrogen and stored at -80 °C until the preformation of RNA isolation.” replace to “The samples were immediately frozen in a liquid nitrogen and then stored at -80 °C until the RNA isolation.”
Response 11: Thank you! According to your suggestion, we have revised it in the line 104-105.
Point 12: Line 104, Pleas provide the performing cDNA protocol and the number of cDNA samples
Response 12: Thank you! We have added “After that, 2μL of total RNA and 3μL of Oligo(dT)18 were mixed at 70°C for denaturation, immediately placed on ice for 5 minutes. Then 7.5μL of ddH2O, 4μL of Reverse Transcription 5×Buffer, 2μL of dNTPs(10mM), 0.5μL of Recombinant RNasin® Ribonuclease Inhibitor and 1μL of M-MLV Reverse Transcriptase were added in each 20μL Reverse Transcription reaction system. After fully mixed, reverse transcription at 42°C for 1 hour, and finally inactivated of reverse transcriptase at 72°C for 10 minutes.” In the line 111-116.
Point 13: Line 120, Please provide the experimental unite
Response 13: The experimental unite we provide is “The data for angel, distance and gene expression were expressed as mean ± standard deviation (M ± SD). We first analyzed the difference of gender (proportion) among left crossed chickens, right crossed chickens and normal chickens by using Chi square test. Since the gender did not affect the type of crossed beak, we then analyzed the difference (proportion) of left-upper cross-beak chickens, left-lower cross-beak chickens, right-upper cross-beak chicken, right-lower cross-beak and polarization cross-beak chicken as well as upper cross-beak chicken and lower cross-beak regardless of gender by Chi square test. Data of gene expression and distance were checked for normality and homogeneity of variance, transformed where necessary. Samples for gene expression were selected for female chickens, which was analyzed using one-way (control, weak, medium and severe) analysis of variance (ANOVA) by SPSS 20.0. While, data of distance were analyzed using two-way analysis of variance (ANOVA) by SPSS 20.0.” In the line130-140.
Point 14: Line 122, Please provide the statistical model
Response 14: Thank you. We re-analyze all the data. With gene expression, samples were taken from female in (n = 6 each group), which is analyzed by one-way ANOVA. Then, the difference of angles of different deformity beak and distances among groups, two factors are considered into account, the gender and groups (control, weak, medium and severe; right crossed beak, left crossed beak, and normal beak). We also provide the model in the statistically section. In the line 140-145.
Point 15: Line 162-163, Please provide reference, if available
Response 15: According to your suggestion, we have added reference [1] in the line 197.
Point 16: Line 163, Replace “make” to “consist”
Response 16: Thank you! According to your suggestion, we have replaced “make” to “consist” in the line 197.
Point 17: Line 175-176, Please discuss with the references used accordingly
Response 17: We are very sorry that we may misunderstand the original meaning of the article. Now, we re-organize the sentence as follow: “These findings indicate that the lower beak is more prone to deformities, which may be supported by a previous study described that the mandible moves easily than other facial bones”. In the line 208-210.
Those suggestions above are necessary and indispensable for us to improve the quality of the paper. Many thanks to you for your good comments.
References:
Bai, H.; Sun, Y.; Zhu, J.; Liu, N.; Li, D.; Xue, F.; Li, Y.; Chen, J., Study on LOC426217 as a candidate gene for beak deformity in chicken. BMC genetics 2016, 17, 44.

Reviewer 2 Report
Summary
The authors investigate the prevalence, sub-type, and relationship of cross-beak in Huiyang bearded chickens. This small work has the potential to be interesting to readers of Animals, however the authors fail to explain why this condition is important. Are Huiyang chickens used extensively in Chinese poultry production, industrialised or among small-stakes farmers? What is the impact of Huiyang’s cross-beak on poultry production in China? The manuscript also falls short in terms of Materials and Methods descriptions, making interpretation of the results difficult.
Major comments
Line 88-91: The authors should comment on the time between decapitation and x-ray. Does rigor mortis of the facial muscles have an effect on beak deviation?
Authors should provide a better example diagram of how x-ray based angles were captured.
How was the interorbital sinus measured? Verbal description and enhanced Figure 1 is necessary. The authors should consider including a mix of both x-rays and schematics to deliver their point.
Lines 97-105: It is unclear what types of tissues the authors were collection from lacrimal, turbinate, etc. Where these dissected bone or chunks of tissue that included both bone and soft tissue?Bone is dense and difficult to homogenise. What method did the authors use to homogenise these tissues? What was the quality of the RNA (RIN values or even exemplary gels would help).
Lines 113-118: Define GADPHby using its full name (see minor comment Line 57 below). The use of RT-PCR controls like GADPHis often debated across species. Did the authors test any other controls to normalise expression?
Line 120-124. Why did the authors choose to use a one-way ANOVA, especially since they appeared to attempt to control for sex in their experimental design?
Lines 147-155: The authors claim that the distances between the left eye to the upper beak and the nostrils is significantly shorter in birds with cross-beak. This might be the case, but I find the use of lateral measurements to be flawed due to parallax. How did the authors control for this phenomena? Or were measurement taken on dead birds with a calliper or other measuring device? The authors need to clearly explain what they did in the materials and methods.
Line 98-99. I could not find any information regarding how many chickens were assessed by qRT-PCR. This information needs to be presented in order to evaluate the meaningfulness of the qRT-PCR experiments.
The percentile summaries listed in Table 2 are confusing. This Table needs to be simplified to indicate how many chickens were assessed overall,the percentile for each dysmorphology by sex. In its current form, it make no sense why the numbers don’t add up to 100%.
The authors are missing an animal ethics statement.
Discussion: The authors would like to conclude that reported upregulation of BMP4expression is potentially causative of cross-beak. This section needs to be toned down. The level of data presented far from making such a conclusion. If anything, BMP4 overexpression is potentially a marker of cross-beak.
Lines 407-409, 439-440: The caption to these figures are unclear (“adjacent letters”). The authors should use asterisks to indicate significance (eg * P<0.05). This caption could also be used as an opportunity to indicate how many animals were used in each experiment. The figure legend in Fig. 3 seems to introduce new classifications of morphology that are not described in the text, especially with regards to the use of “weak”, “medium”, and “severe”.
Minor comments
Line 36: From its description, cross-beak is probably better described as a “dysmorphology” than a “disease”. The former can lead to disease.
Line 20: “hasnegative” à“has negative”
Line 51: “traumata” à“trauma”
Line 57: Use gene’s full name in first instance of its abbreviation (eg “…bone morphogenetic protein 4 (BMP4)…”)
Line 112-113: Unnecessary break in paragraph.
Line 191-194: “One intriguing possibility was that during embryonic 191 development, some craniofacial bones are slightly distorted, leading to the generation of 192 cross-beaks. The other explanation may be due to some craniofacial bones are slightly distorted 193 with the generation of cross-beaks.” These two sentences are redundant as written.
The authors need to revisit the journal’s reference format, as there appear to be inconsistencies in the current manuscript.
Author Response
Dear Editors and Reviewers:
Thank you for your letter and for the reviewers’ comments concerning our manuscript entitled “animals-631631”. Those comments are all valuable and very helpful for revising and improving our paper, as well as the important guiding significance to our researches. We have studied comments carefully and have made correction which we hope meet with approval. Revised portion are marked in red in the paper. The main corrections in the paper and the responds to the reviewer’s comments are as flowing:
Responds to the Reviewers’ comments:
Point 1: Line 88-91: The authors should comment on the time between decapitation and x-ray. Does rigor mortis of the facial muscles have an effect on beak deviation?
Response 1: Thanks. X-ray was implemented within 6 hours after decapitation, and we measured the deviated angel immediately after chickens being slaughtered. We think very limited change of the facial muscles occurs, and all samples were treated under the same protocol.
Point 2: Authors should provide a better example diagram of how x-ray based angles were captured.
Response 2: Thanks. The x-ray was not used to measure angels or distance in the our study, but to capture pictures displayed in Figure 1 to show an example of deformity beak, which is not fixed based on certain angles.
Point 3:How was the interorbital sinus measured? Verbal description and enhanced Figure 1 is necessary. The authors should consider including a mix of both x-rays and schematics to deliver their point.
Response 3: Thanks. We cut along the yellow dotted line to expose the infraorbital sinus. The length of left/right infraorbital sinus (LIS/RIS) were measured with a vernier caliper. The yellow dotted line is vertically divided equally between the line connecting the leading edge of the corner of the eye and the commissure of lips. We added the verbal description and enhanced the Figure 1 in the line 415-419. We also provide a schematic diagram of the specific location of the infraorbital sinus, as follows.
Point 4:Lines 97-105: It is unclear what types of tissues the authors were collection from lacrimal, turbinate, etc. Where these dissected bone or chunks of tissue that included both bone and soft tissue? Bone is dense and difficult to homogenise. What method did the authors use to homogenise these tissues? What was the quality of the RNA (RIN values or even exemplary gels would help).
Response 4: Thanks. The samples we used were bone tissue, and we had removed as possible soft tissue as we could, including periosteum. And we have added “The bone tissues were placed in a beaded tube (SC Micro Tube, Sarstedt) with Trizol reagent (1 ml/100 mg of tissue). The FastPrep-24™5G (MP Biomedicals) was used to agitate the sample tube at a speed of 5 m/s for 35 s. And then, samples were centrifuged at 12,000 rpm for 15 min at room temperature and the upper aqueous layer was transferred to a new tube for next RNA extraction.” in the line 105-109. We didn’t measure the RIN values, but the PCR products were detected by 1 % agarose gel electrophoresis, the agarose gel electrophoretogram is as follow.
Agarose gel electrophoretogram for RT-PCR products of GAPDH and BMP4 gene.
Point 5:Lines 113-118: Define GADPH by using its full name (see minor comment Line 57 below). The use of RT-PCR controls like GADPH is often debated across species. Did the authors test any other controls to normalise expression?
Response 5: Thanks.We tested GAPDH and β-actin genes by using geNorm and Normfinder software, and the result showed that the GADPH was the most stable gene.
Point 6:Line 120-124. Why did the authors choose to use a one-way ANOVA, especially since they appeared to attempt to control for sex in their experimental design?
Response 6: Thank you! We previously considered the cross-beaks to be an autosomal inherited disease, so we ignored the gender factor. Now, we re-analyze all the data. With gene expression, samples were taken from female in (n=6 each group), which is analyzed by one-way ANOVA. Then, the difference of angles of different deformity beak and distances among groups, two factors are considered into account, the gender and groups (control, weak, medium and severe; right crossed beak, left crossed beak, and normal beak). We also provide the model in the statiscally section. In the line 130-145.
Point 7:Lines 147-155: The authors claim that the distances between the left eye to the upper beak and the nostrils is significantly shorter in birds with cross-beak. This might be the case, but I find the use of lateral measurements to be flawed due to parallax. How did the authors control for this phenomena? Or were measurement taken on dead birds with a calliper or other measuring device? The authors need to clearly explain what they did in the materials and methods.
Response 7: Thanks. We measured the distance of eye (midpoint) to that of nostrils with the vernier caliper, and we considered all the distance are obtained under the same condition.
Point 8:Line 98-99. I could not find any information regarding how many chickens were assessed by qRT-PCR. This information needs to be presented in order to evaluate the meaningfulness of the qRT-PCR experiments.
Response 8: Thanks. Total 24 chickens (n=6 in each group, weak, medium, severe and control) were measured for RNA expression, which was written in the line 103-104 .
Point 9:The percentile summaries listed in Table 2 are confusing. This Table needs to be simplified to indicate how many chickens were assessed overall, the percentile for each dysmorphology by sex. In its current form, it make no sense why the numbers don’t add up to 100%.
Response 9: Thank you! According to your suggestion, we have simplified the Table 2 in the line 462-464. In the previous table, all the cross-beak = Left-upper cross-beak + Left-lower cross-beak + Right-upper cross-beak + Right-lower cross-beak + Polarization cross-beak, the proportion of these five types of cross-beak are add up to 100%. Otherwise, all the cross-beak = Upper cross-beak(Left-upper cross-beak + Right-upper cross-beak) + Lower cross-beak(Left-lower cross-beak + Right-lower cross-beak) + Polarization cross-beak, the proportion of these three types of cross-beak are add up to 100%.
Point 10:Discussion: The authors would like to conclude that reported upregulation of BMP4expression is potentially causative of cross-beak. This section needs to be toned down. The level of data presented far from making such a conclusion. If anything, BMP4 overexpression is potentially a marker of cross-beak.
Response 10: Thank you. We may overstate the conclusion and re-organize that “This finding provides an insight of a potential role of BMP4 in the development for cross-beaks.” in line 33-34 and line 276.
Point 11:Lines 407-409, 439-440: The caption to these figures are unclear (“adjacent letters”). The authors should use asterisks to indicate significance (eg * P<0.05). This caption could also be used as an opportunity to indicate how many animals were used in each experiment. The figure legend in Fig. 3 seems to introduce new classifications of morphology that are not described in the text, especially with regards to the use of “weak”, “medium”, and “severe”.
Response 11: According to your suggestion, we use asterisks to indicate significance and remade the figure and table. Total 24 chickens (n=6 in each group, weak, medium, severe and control) were measured for RNA expression, which was written in the line 103-104. We introduced classifications for “weak”, “medium”, and “severe cross-beak” chickens in the line 94-95.
Point 12:Line 36: From its description, cross-beak is probably better described as a “dysmorphology” than a “disease”. The former can lead to disease.
Response 12: Thank you. We have replaced the word “disease” to “dysmorphology” in the line 38 and line 42.
Point 13:Line 20: “hasnegative” à “has negative”.
Response 13: We are very sorry for such a mistake. We have revised it in the line 20-21.
Point 14:Line 51: “traumata” à “trauma”
Response 14: Thank you! According to your suggestion, we have replaced “traumata” to “trauma”. In the line 53.
Point 15:Line 57: Use gene’s full name in first instance of its abbreviation (eg “…bone morphogenetic protein 4 (BMP4)…”)
Response 15: Thank you! According to your suggestion, we have supplemented the expansion of “BMP4” in line 12, line 24 and line 58.
Point 16:Line 112-113: Unnecessary break in paragraph.
Response 16: We are very sorry for such a mistake. We have revised it.
Point 17:Line 191-194: “One intriguing possibility was that during embryonic 191 development, some craniofacial bones are slightly distorted, leading to the generation of 192 cross-beaks. The other explanation may be due to some craniofacial bones are slightly distorted 193 with the generation of cross-beaks.” These two sentences are redundant as written.
Response 17: Thank you for your suggestion. After our discussion, we agreed with you very much and have deleted these two sentences. In the line 226.
Point 18:The authors need to revisit the journal’s reference format, as there appear to be inconsistencies in the current manuscript.
Response 18: Thank you! We have rechecked the reference format and revised them accordingly.
Those suggestions above are necessary and indispensable for us to improve the quality of the paper. Many thanks to you for your good comments.

Round 2
Reviewer 1 Report
The authors revised the manuscript as proposed, the manuscript is now accepted for publication.
Reviewer 2 Report
No additional comments. The manuscript changes are sufficient to merit acceptance for publication.